# Data Similarity is Not Enough to Explain Language Model Performance

**Gregory Yauney**[1*]    **Emily Reif**[2]    **David Mimno**[1]
[1]Cornell University    [2]Google Research
gyauney@cs.cornell.edu    ereif@google.com    mimno@cornell.edu

## Abstract

Large language models achieve high performance on many but not all downstream tasks. The interaction between pretraining data and task data is commonly assumed to determine this variance: a task with data that is more similar to a model's pretraining data is assumed to be easier for that model. We test whether distributional and example-specific similarity measures (embedding-, token- and model-based) correlate with language model performance through a large-scale comparison of the Pile and C4 pretraining datasets with downstream benchmarks. Similarity correlates with performance for multilingual datasets, but in other benchmarks, we surprisingly find that similarity metrics are not correlated with accuracy or even each other. This suggests that the relationship between pretraining data and downstream tasks is more complex than often assumed.

## 1 Introduction

Large language models (LMs) are pretrained on large datasets, such as C4 (Raffel et al., 2020) and the Pile (Gao et al., 2020), and used for downstream tasks. Many hypothesize that task performance is a matter of how similar the task data is to the pretraining data (Brown et al., 2020; Gao et al., 2020; Gonen et al., 2022). Despite the prevalence of this "similarity hypothesis," it has not yet been evaluated beyond data overlap (Kandpal et al., 2023). Can we measure the impact of data similarity?

Similarity matters at a coarse level: models pretrained on English data do not perform as well on other languages (Bender, 2019; Scao et al., 2022). Robustness to distribution shifts has been extensively studied (Hendrycks et al., 2020; Koh et al., 2021), though dissimilarity is usually assumed through dataset construction rather than explicitly measured. We measure distributional similarity between pretraining datasets and benchmarks using a

---

* Work done as a Student Researcher at Google Research.

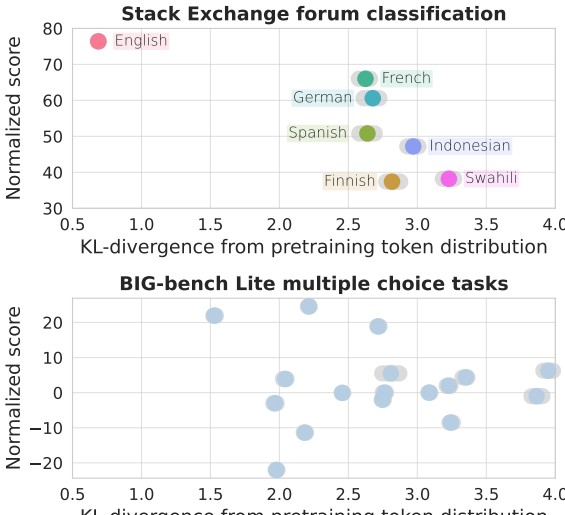

Figure 1: Similarity to the pretraining dataset correlates with Pythia-6.9B zero-shot performance when comparing different-language versions of a single task (top), but similarity fails to explain performance differences in BIG-bench Lite (bottom). Error bars are over samples of the pretraining dataset.

range of similarity metrics, finding that similarity correlates with performance in a controlled setting but not across common benchmarks (Figure 1).

Example-level similarity offers a more targeted hypothesis that controls for task difficulty: within a downstream dataset, examples that are similar to examples in the pretraining data should be easier than dissimilar examples. It has been shown that examples with words and entities more frequently found in the pretraining dataset (e.g. Obama or the number 24) are more often answered correctly in arithmetic and factoid QA tasks (Razeghi et al., 2022; Kandpal et al., 2023; Biderman et al., 2023). We ask a stronger version of this question: will a model perform well on an example if that example is similar to *any* pretraining document?

We test the hypotheses that distributional similarity and example-similarity determine performance

using three measurements of similarity from prior work: word similarity, embedding similarity, and language model perplexity. Our contributions are the following. First, we show that similarity correlates with performance for multilingual datasets. The same underlying task is represented in different languages, each with a different aggregate similarity to the pretraining dataset, and languages that are more similar to the English pretraining data afford higher performance. Second, we show that multiple measures of distributional similarity are surprisingly not correlated with zero-shot performance on BIG-bench Lite multiple-choice tasks (Srivastava et al., 2023) across models and pretraining datasets. Third, we show that within a downstream dataset, examples that are more similar to any example in the pretraining dataset are not easier than less-similar examples. We do this by comparing examples from GLUE (Wang et al., 2019) and BIG-bench Lite with *all* of C4 and the Pile. Fourth, we find that the similarity measures are not even correlated with each other. We conclude with several examples that show the limits of similarity's predictive power. Although similarity is a common post-hoc justification for performance, we find it is misleading. This indicates the importance of model-based interpretability approaches, like training data attribution (Akyürek et al., 2022).[1]

## 2   Related work

The impact of pretraining data on downstream performance is an active area of research. Similarity to a model's pretraining dataset is assumed to determine the model's performance on tasks (Radford et al., 2019; Brown et al., 2020; Gonen et al., 2022). Prior work has focused on repeated entities in arithmetic (Razeghi et al., 2022) and QA tasks (Liu et al., 2022; Kandpal et al., 2023). Finetuned downstream performance benefits from continued pretraining on a similar domain (Gururangan et al., 2020; Xie et al., 2023).[2] Contemporaneous work quantifies task similarity to downstream training datasets (Yuan et al., 2023; Li et al., 2023). Task data leakage is an extreme case of data similarity, where the presence of task examples in the pretraining dataset impacts downstream performance on that task (Dodge et al., 2021; Magar and Schwartz, 2022). LM perplexity has been assumed to be

a proxy for similarity to pretraining data (Gonen et al., 2022), though it is poorly calibrated for downstream tasks (Zhao et al., 2021; Ren et al., 2022). Training data attribution uses influence functions, gradients, and model internals rather than data similarity (Pruthi et al., 2020; Akyürek et al., 2022; Han and Tsvetkov, 2022). Other approaches quantify task difficulty but do not explicitly account for the role of pretraining data (Ethayarajh et al., 2022; Yauney and Mimno, 2021; Swayamdipta et al., 2020; Le Bras et al., 2020; Perez et al., 2021).

## 3   Similarity correlates with performance in a controlled setting

The similarity hypothesis implies that a downstream dataset's similarity to the pretraining data should matter. As an initial check of this hypothesis, we construct multiple versions of a task. The underlying task is held constant, but we decrease the similarity of the data to the training set in a way that we expect to decrease LM performance. Datasets in multiple languages are a starting point because text in non-English languages is presumably less similar to English pretraining data. While popular English pretraining datasets do contain non-English text that enables some transfer to other languages, LMs trained on English do not perform as well on multilingual tasks (Blevins and Zettlemoyer, 2022; Lin et al., 2022; Gao et al., 2020).

We consider two tasks that have datasets in English and several other languages, where the task is fixed across languages. For an explicitly easy task, we extend a Stack Exchange forum classification dataset (Yauney and Mimno, 2021) by translating it from English into multiple languages. For a harder task, we use XNLI (Conneau et al., 2018). Details are in Appendix A. We measure zero-shot performance of Pythia-6.9B (Biderman et al., 2023), which was pretrained on the Pile (Gao et al., 2020), a large dataset of mostly English text.

Figure 1 (top) shows that performance tracks the KL-divergence of unigram token distributions between the Pile and Stack Exchange datasets. The English dataset is the most similar dataset to the Pile and affords the highest performance. All other languages are much less similar to the Pile and have lower performances. Note that token distribution similarity does not allow fine-grained comparisons between the languages with high KL-divergence.

We also test whether performance smoothly degrades as we titrate in a given amount of dissimilar-

---

[1]Code and data are available at: https://github.com/gyauney/data-similarity-is-not-enough

[2]We find finetuned performance does not correlate with vocabulary overlap for the data from Gururangan et al. (2020).

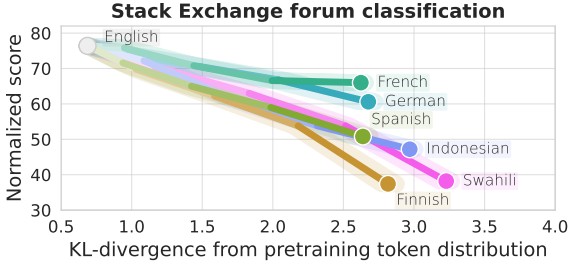

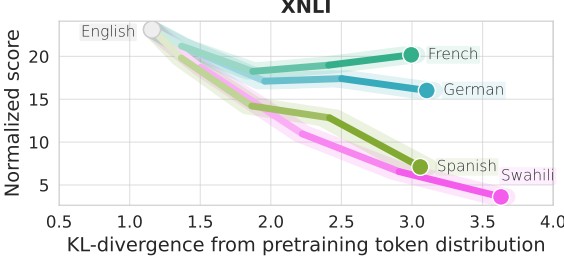

Figure 2: Mixed-language datasets show that performance smoothly decays as more of the dataset is translated from English into another language. Each line represents a dataset that begins as 100% English, with an increasing percentage of each document translated into another language, until completely translated. Error bars are over samples of the pretraining dataset.

ity. We construct datasets that interpolate between English and a translation by translating an increasing percentage of each example. Figure 2 shows that decreasing similarity tends to decrease performance for Stack Exchange datasets and the XNLI languages that are in the Latin alphabet. Figure 5 in the appendix shows the same trend for all of XNLI.

## 4 Measuring similarity

Having established that distributional similarity correlates with performance in a setting that holds task difficulty constant, can we say the same across different benchmark tasks? And is performance higher for more similar examples? If the similarity hypothesis holds more generally, so that similarity to training examples can consistently predict performance on tasks, we expect that multiple measures of similarity will correlate with accuracy.

**Similarity measures.** We implement three categories of similarity measures on different text representations: First, KL-divergence of $n$-gram distributions. Second, embedding-based similarities using Sentence-T5 embeddings (Ni et al., 2022), including: (a) MAUVE score (Pillutla et al., 2021) of a downstream dataset against samples of the pretraining dataset, (b) the maximum cosine similarity between a downstream example and the *entire* pre-

| | Pythia-6.9B | T5 v1.1 XL | | Flan-T5 XL | |
|---|---|---|---|---|---|
| | 0-shot | 0-shot | 2-shot | 0-shot | 2-shot |
| Bigram KL-divergence ($-$) | $-0.06$ 0.837 | $-0.10$ 0.708 | $-0.04$ 0.897 | $-0.67$ 0.005 | $-0.64$ 0.007 |
| MAUVE score ($+$) | $-0.36$ 0.165 | $0.19$ 0.478 | $0.27$ 0.318 | $0.26$ 0.329 | $0.46$ 0.075 |
| Max cosine similarity ($+$) | $0.08$ 0.778 | $-0.10$ 0.716 | $-0.26$ 0.339 | $0.14$ 0.594 | $-0.13$ 0.633 |
| Mean cosine similarity ($+$) | $0.07$ 0.795 | $-0.05$ 0.867 | $-0.19$ 0.478 | $0.19$ 0.492 | $-0.10$ 0.713 |
| Input perplexity ($-$) | $0.09$ 0.729 | $-0.13$ 0.644 | $-0.24$ 0.374 | $-0.16$ 0.542 | $0.12$ 0.664 |
| Correct target perplexity ($-$) | $0.44$ 0.085 | $0.09$ 0.745 | $-0.02$ 0.948 | $-0.21$ 0.431 | $-0.25$ 0.356 |

Table 1: Spearman correlation coefficients between aggregate similarity measures and normalized score for BIG-bench Lite at the task level. $p$-values are in small font. $+/-$ next to similarity shows whether more similar datasets have higher/lower values. No correlation is significant at $p < 0.0017$ after Bonferroni correction.

training dataset, and (c) the mean cosine similarity between a downstream example and the closest 1,000 neighbors in the *entire* pretraining set. Third, language modeling loss of a pretrained model evaluated on a downstream example. We use the perplexity of the input sequence and the perplexity of the correct target conditioned on the input.

We further measure similarity at two different scales: 1) aggregate similarity between an entire downstream dataset and the pretraining dataset, and 2) example-level similarity between an individual downstream example and the pretraining dataset. Cosine similarities and LM loss are example-level, so we take the mean over a dataset's examples for aggregate similarity. See Appendix B for details.

**Models and datasets.** We use models pretrained on two datasets: C4 (Raffel et al., 2020; Dodge et al., 2021) and the Pile (Gao et al., 2020). We use T5 v1.1 XL (Raffel et al., 2020), which was pretrained solely on C4, and Flan-T5 XL (Chung et al., 2022), which was further instruction-tuned on over 1,800 downstream tasks (Longpre et al., 2023a). For the Pile, we use Pythia-6.9B (Biderman et al., 2023). Datasets and models are from the `transformers` and `datasets` libraries (Wolf et al., 2020; Lhoest et al., 2021).

We test our hypotheses on GLUE tasks (Wang et al., 2019) and multiple-choice tasks from the more difficult BIG-bench Lite suite (Srivastava et al., 2023). We formulate tasks using finetuning for GLUE.[3] For BIG-bench Lite, we use in-context learning, which does not require any parameter updates. Following Srivastava et al. (2023), we normalize performance across tasks that have different random baselines by rescaling accuracy to 0 for random chance and 100 for perfect. See Appendix C for details.

---

[3]Pretrained-only performance is no better than random.

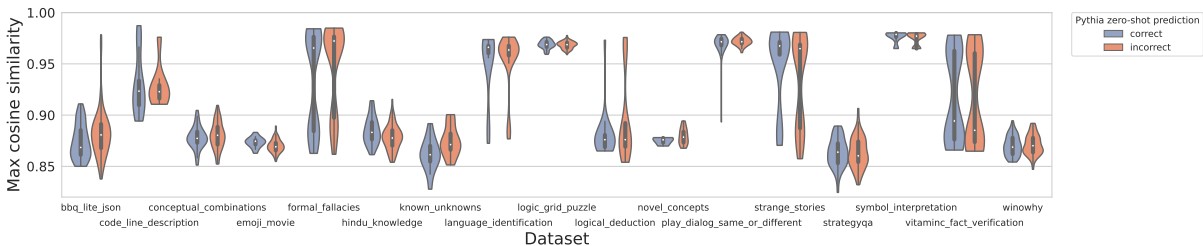

Figure 3: Correctly-classified examples from BIG-bench are not consistently more similar to pretraining data based on Pythia-6.9B zero-shot performance and each example's maximum cosine similarity to any document in the Pile.

## 5 Results

**Tasks that are more similar to the pretraining data do not have higher few-shot performance.** We first use BIG-bench Lite to examine whether datasets that are more similar in aggregate to the pretraining data are easier for a given model. If the similarity hypothesis is true, we expect to see significant high-magnitude correlations between performances and similarities to the corresponding pretraining dataset. Table 1 shows the lack of correlation between five aggregate similarity measures and few-shot performance on BIG-bench Lite tasks. No correlation is statistically significant at the level of $p < 0.0017$ after Bonferroni correction for multiple tests (Dror et al., 2017).[4] The strongest correlations are for KL-divergence and MAUVE with performance of Flan-T5 XL, the best performing model. If these correlations were to hold in a more targeted setting, then text-based similarities would be more promising than the embedding and model-based similarities we study. Appendix D discusses statistical considerations.

**Examples that are more similar to any pretraining document are not easier.** We can control for different task difficulties by focusing on individual examples within a dataset. If the example-level similarity hypothesis is true, then we expect to see that within a task, examples that are more similar to the pretraining dataset are classified correctly more frequently than less similar examples. Figure 3 shows that BIG-bench Lite examples classified correctly by Pythia-6.9B do not have significantly higher embedding cosine similarity to any document in the Pile than those that were misclassified. This holds for all the tasks we examined. Figure 6 in the appendix shows that accuracy of more similar examples is not consistently higher than accuracy of dissimilar examples. Figure 7 in the appendix

---

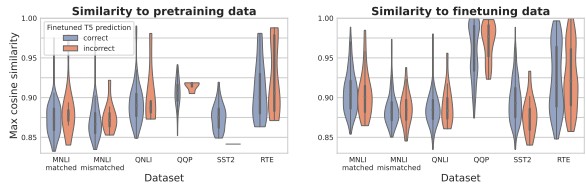

Figure 4: Finetuned T5 3B's performance on GLUE validation examples does not correlate with embedding similarity to the pretraining dataset (left) or similarity to the finetuning training dataset (right).

shows this result for T5 performance and maximum similarity to C4 documents.

We also test the hypothesis in the finetuning paradigm by using embedding similarity and T5 3B finetuned on GLUE. Here we compare examples from the validation split to both the pretraining dataset and the GLUE training split. Figure 4 shows no correlation between performance on an example and that example's maximum similarity to documents in the pretraining or finetuning datasets.

## 6 Discussion

Our goal is to evaluate the similarity hypothesis: that tasks that are more similar to pretraining data are easier. We have so far shown that similarity does not in practice correlate with performance for common benchmark tasks. In this process we found additional reasons to doubt the hypothesis.

**Task difficulty can be orthogonal to similarity.** Tasks can have language similar to the pretraining corpus and still be hard. To underscore that similarity is not sufficient to predict performance, we use another Stack Exchange classification task. Rather than predicting whether a given post was from the bicycles or cstheory forum, we instead choose a much more difficult task: predicting whether it was posted before or after noon. Again, we translate the dataset into a range of other languages. By construction, the similarity to the pretraining data

---

[4]The probability of any false positives is at most 0.05.

is nearly the same as in Figure 1, but the accuracy is random (Figure 8, Appendix).

**Similarity measures are uncorrelated.** Do different similarity measures capture the same notion of similarity? Surprisingly, different measures of aggregate similarity between downstream datasets and pretraining datasets are not significantly correlated with each other when comparing BIG-bench Lite tasks with C4 and the Pile (Table 2). These results indicate that similarity is not a monolithic concept. Future work could explore a) the affordances of these different kinds of similarities and b) whether more refined similarity measures might better correlate with performance.

**Scope of our findings.** We do not disprove the similarity hypothesis. One possibility is that all datasets in BIG-bench Lite and GLUE are so similar to C4 and the Pile that similarity differences are not great enough to impact performance. Our experiments are on models up to the 7B-parameter scale. Perhaps there is a model scale threshold at which similarity matters more for performance. It is clear that at least some level of performance variation is necessary to see if similarity matters. For example, a model that performs poorly on all tasks (or conversely if it solves all tasks perfectly) will not be able to show any meaningful influence of similarity. And while we evaluate a broad range of similarity measures, they do not capture all possible similarities. Rather, our results make it increasingly unlikely that similarity—broadly construed—is the single most determinative factor in this setting.

## 7 Conclusion

At a broad level, it is clear that pretraining data matters. Data curation can impact robustness, memorization, and toxicity generation (Nguyen et al., 2022; Lee et al., 2022; Longpre et al., 2023b). Specific examples of high-level task similarity, such as prevalence of entity mentions, have been observed to correlate with performance on arithmetic and QA tasks (Razeghi et al., 2022; Kandpal et al., 2023). We find that while similarity is correlated with performance when task difficulty can be kept constant, it is surprisingly hard to connect textual similarity to performance for standard benchmarks in the few-shot setting. Both distributional and fine-grained example-based similarity to the pretraining dataset are not enough to explain performance across benchmarks, and this remains true for every

| C4 | | | | |
|---|---|---|---|---|
| | KL-divergence | MAUVE score | Max cosine similarity | Input perplexity |
| MAUVE score | −0.36 0.165 | | | |
| Max cosine similarity | −0.24 0.380 | −0.48 0.059 | | |
| Input perplexity | −0.19 0.478 | 0.26 0.338 | −0.29 0.284 | |
| Correct target perplexity | −0.42 0.107 | 0.30 0.263 | −0.07 0.795 | 0.08 0.762 |

| The Pile | | | | |
|---|---|---|---|---|
| | KL-divergence | MAUVE score | Max cosine similarity | Input perplexity |
| MAUVE score | −0.41 0.117 | | | |
| Max cosine similarity | −0.34 0.204 | −0.38 0.152 | | |
| Input perplexity | −0.11 0.696 | 0.28 0.302 | −0.55 0.028 | |
| Correct target perplexity | −0.24 0.362 | −0.07 0.803 | 0.32 0.231 | 0.04 0.888 |

Table 2: Aggregate similarity measures are not significantly correlated with each other when comparing BIG-bench Lite datasets with C4 and the Pile. Perplexities are from T5 v1.1 XL and Pythia-6.9B, respectively. Spearman correlation coefficients with $p$-values.

model and operationalization of similarity that we tried. While theoretical work suggests that the similarity between training and test distributions plays a role in upper-bounding error for LMs applied to downstream classification tasks (Saunshi et al., 2021) and in more general settings (McNamara and Balcan, 2017), we find that "similarity" is surprisingly difficult to measure in practice and that other factors may have more impact on difficulty.

**Future work.** Future work can study the advantages of training data attribution over raw similarity, as begun by Mozes et al. (2023). Pretraining on datasets constructed to be more or less similar to specific downstream tasks might also be able to determine if there is a causal relationship between similarity and performance.

## Limitations

Embedding comparisons between downstream datasets and pretraining datasets are computationally expensive due to the large size of the latter. We are therefore limited in how many downstream datasets we can compare to pretraining datasets. We strike a balance by working with previously-studied similarity measures and standard diverse benchmarks like BIG-bench, though this topic would benefit from additional datasets and models. We must control for multiple comparisons when testing the significance of correlations, reducing the statistical power of our experiments. More targeted work could investigate specific kinds of similarity. Our experiments focus on the few-shot setting, and our results are not necessarily valid for all settings. Although our main results contradict conventional wisdom in this setting, we have performed this work in order to provide a more solid foundation for new pretraining data analyses.

## Ethics statement

We do not anticipate any harms from our use of publicly available datasets. While large language models are carbon-expensive to pretrain, the few-shot paradigm that we focus on does not require additional compute-intensive parameter updates.

## Acknowledgments

We thank Daphne Ippolito, Katherine Lee, Dheeraj Rajagopal, and Lisa Schut for discussions and technical help; Daniel Smilkov and Nikhil Thorat for help scaling embeddings to full datasets; and Lucas Dixon, Rebecca Hicke, Andrea Wang, Matthew Wilkens, and Ann Yuan for writing feedback. We also thank the anonymous reviewers.

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

## A  Dataset details

**Stack Exchange.** The Stack Exchange forum identification task provides an easy task (Yauney and Mimno, 2021). It consists of 1,000 English-language posts, and the task is to predict whether a post is from the bicycles or cstheory forum. We translate the posts into multiple languages using Google Translate. It also includes AM/PM labels for a much more difficult task.

**XNLI.** XNLI is a natural language inference task that also comes in many languages and is much more difficult in the zero-shot setting (Conneau et al., 2018). We use the validation set of 2,490 documents in each language. We construct intermediate translations by translating the first $\{25\%, 50\%, 75\%\}$ of each example in English XNLI into the other XNLI languages using Google Translate. Figure 2 shows languages that are well-tokenized by the Pythia tokenizer, i.e. those in the

Latin alphabet. Figure 5 shows qualitatively similar results for all languages.

**BIG-bench Lite.** BIG-bench Lite is a subset of diverse tasks from the BIG-bench suite (Srivastava et al., 2023). It consists of 18 multiple-choice and 6 text generation tasks. We report results using the multiple-choice tasks. We exclude the generative tasks from our analysis because both Pythia-6.9B and T5 v1.1 XL do not succeed beyond 0 performance on any of them. Results for the generative tasks are included in the code repository. We exclude two tasks, misconceptions_russian and parsinlu_reading_comprehension, that cannot be tokenized by the T5 tokenizer. We also exclude winowhy from aggregate analysis.

**GLUE.** We evaluate English-language datasets from GLUE (Wang et al., 2019), with an emphasis on natural language inference. We use the validation splits of MNLI, QNLI, QQP, SST2, and RTE.

## B  Similarity metric details

$n$**-gram baselines.** We represent a dataset as a distribution over tokens and calculate the KL-divergence between a pretraining dataset and each downstream dataset. We represent text as bigrams hashed to a 10,000-dimensional vector, as in Xie et al. (2023), and by unigrams from the Pythia tokenizer. We sample 100,000 documents at a time from the pretraining dataset and compare against all of each downstream dataset.

**Cosine similarity.** We represent documents using Sentence-T5 embeddings (Ni et al., 2022). For a sample of documents in a downstream dataset, we calculate a) the maximum cosine similarity between a downstream document and the entire pretraining dataset and b) the mean cosine similarity between a downstream document and the closest 1,000 neighbors in the entire pretraining dataset. For GLUE, we sample 250 documents from each task. We sample up to 100 documents from each BIG-bench Lite task (some tasks have fewer than 100 documents). We observed similar results with BERT embeddings (Devlin et al., 2019). We find that the two variants of cosine similarity are correlated ($\rho = 0.95$, $p < 10^{-4}$), as is to be expected. Max cosine similarity can measure task data leakage: since downstream examples are compared to all pretraining documents, exact matches will have a maximum cosine similarity of 1.

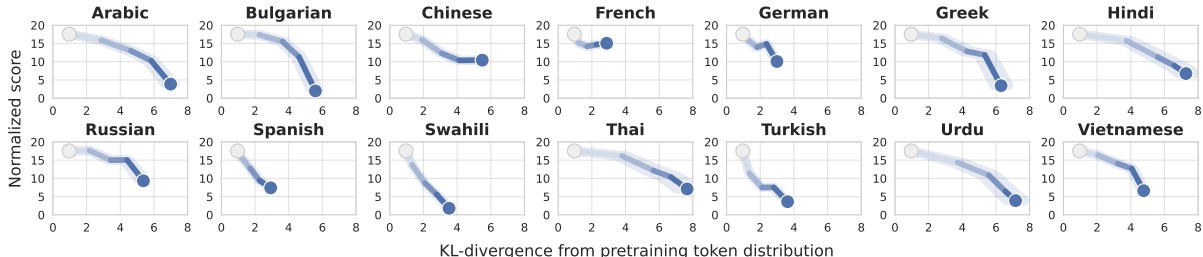

Figure 5: Full XNLI results for all languages show the trend seen in Figure 2 for Latin-alphabet languages. Each line represents a dataset that begins as 100% English (gray dot in the upper left), with an increasing percentage of each document translated into another language, until completely translated (blue dot in the lower right).

**MAUVE score.** MAUVE is a recent text similarity measure that treats a text dataset as a distribution over embedding space (Pillutla et al., 2021). We again represent documents using Sentence-T5 embeddings and calculate average MAUVE scores between each downstream dataset and 10,000-document samples from the pretraining dataset.

**Perplexity.** LM log-loss per target token (which is log-perplexity for autoregressive models) on the pretraining validation set has been theoretically shown to impact downstream performance (Saunshi et al., 2021). We use log-loss per token on the downstream text as a possible proxy for similarity to the pretraining data, as in Gonen et al. (2022). We use the mean across examples of *input perplexity* and *correct target perplexity*, i.e. the perplexity of the correct target conditioned on the input.

## C In-context learning details

In-context learning, also called the zero-shot or few-shot setting, formats classification tasks as natural language questions (Brown et al., 2020; Dong et al., 2022). A task is defined by a prompt that includes several parts: an instance-specific text input $\mathbf{x}$, an optional instruction $\mathbf{i}$ that provides more context for the desired output, and a closed set of targets $\mathcal{T}$ that defines the possible choices. The predicted choice $\hat{y}$ is the target with the highest probability: $\hat{y} = \arg\max_{\mathbf{t} \in \mathcal{T}} p_\theta(\mathbf{t} \mid \mathbf{x} \circ \mathbf{i})$, where $\circ$ is the concatenation operator and $p_\theta$ is the conditional distribution over words for the model with parameters $\theta$.[5] A few-shot prompt includes one or more examples of desired input/output tuples that are prepended to the input. We do not run any finetuning and instead use the T5 3B model, which was finetuned on GLUE. Prompts and raw results are included in the code repository.

[5]Other scoring methods yield qualitatively similar results.

| | Pythia-6.9B | T5 v1.1 XL | | Flan-T5 XL | |
|---|---|---|---|---|---|
| | 0-shot | 0-shot | 2-shot | 0-shot | 2-shot |
| Bigram KL-divergence (−) | −0.09 0.754 | 0.05 0.841 | 0.02 0.942 | −0.55 0.029 | −0.64 0.007 |
| MAUVE score (+) | −0.26 0.331 | −0.14 0.616 | −0.04 0.873 | 0.40 0.122 | 0.32 0.226 |
| Max cosine similarity (+) | 0.39 0.132 | −0.21 0.435 | −0.32 0.227 | 0.05 0.840 | −0.14 0.595 |
| Mean cosine similarity (+) | 0.41 0.119 | −0.19 0.472 | −0.29 0.284 | 0.08 0.771 | −0.13 0.632 |
| Input perplexity (−) | −0.06 0.835 | 0.08 0.775 | 0.04 0.888 | −0.16 0.545 | 0.09 0.733 |
| Correct target perplexity (−) | 0.58 0.018 | −0.04 0.869 | 0.06 0.838 | −0.19 0.492 | −0.16 0.545 |

Table 3: Pearson correlation coefficients between aggregate similarity measures and normalized score for BIG-bench Lite at the task level. $p$-values are in small font. No correlation is significant at $p < 0.0017$ after Bonferroni correction.

## D Statistical considerations

**Multiple comparisons.** We seek to test the correlation between multiple measures of similarity and multiple performance measures. Using a significance value of $p < 0.05$ for rejecting null hypotheses would give a large overall false positive rate. We use Bonferroni correction, a standard method for controlling for multiple comparisons (Dror et al., 2017). However, Bonferroni correction has been found to decrease statistical power, i.e. it makes false negative results more likely. More complicated corrections that do not sacrifice as much power, like the Benjamini–Yekutieli procedure for possibly dependent hypotheses (Benjamini and Yekutieli, 2001), still do not yield significant correlations (see code for details). Future empirical work would benefit from studying more datasets in order to increase the statistical power of experiments (Card et al., 2020). The code also includes equivalent analysis using $p$-values calculated from permutation tests.

**Pearson correlation.** In the main paper, we report Spearman rank correlation, which measures the degree to which variables are related by a (possibly nonlinear) monotonic function. Pearson correlation is also commonly used to quantify linear

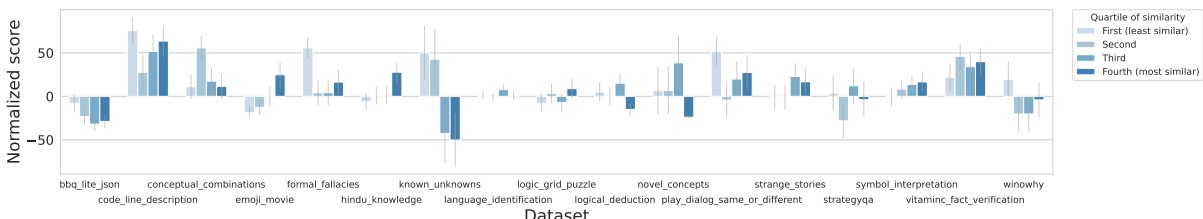

Figure 6: BIG-bench Lite examples that are more similar to pretraining data do not consistently have higher accuracy than less similar examples based on Pythia-6.9B zero-shot performance and each example's maximum cosine similarity to any document in the Pile.

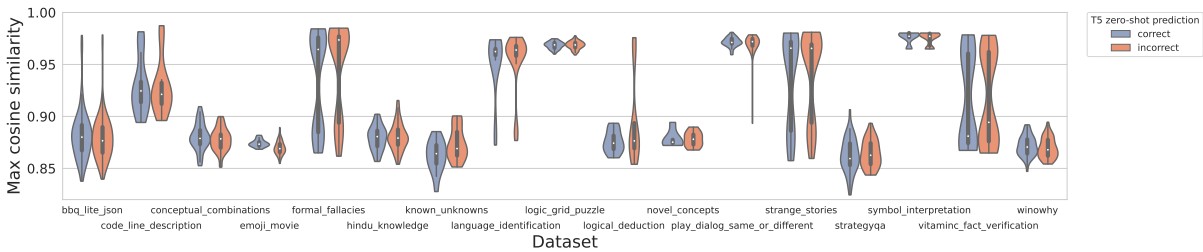

Figure 7: Correctly-classified examples from BIG-bench are not consistently more similar to pretraining data based on T5 v1.1 XL zero-shot performance and each example's maximum cosine similarity to any document in C4.

relationships. Table 3 shows Pearson correlation coefficients for the same comparisons as in Table 1. As before, none of these are significant after Bonferroni correction or the Benjamini-Yekutieli procedure.

## E   Infrastructure and runtime

Zero-shot experiments were run on an Intel Xeon CPU E5-2665 @ 2.40GHz with 250 GB of RAM and an NVIDIA RTX A6000 GPU. Runtimes for BIG-bench Lite tasks ranged from 2 seconds for auto_debugging to 6.3 hours for vitaminc_fact_verification. The whole benchmark suite took 17.1 hours. Evaluating Stack Exchange datasets took 18.2 minutes on average for a total of 7.5 hours. Evaluating XNLI took an average of 10.4 minutes per run for a total of 2.9 hours for Latin-alphabet datasets and 8.7 hours for all datasets under various degrees of translation. Evaluating GLUE took from 4.3 minutes for RTE to 9.3 hours for QNLI, for a total of 18.3 hours.

Calculating embedding cosine similarities between a downstream document and the entirety of a pretraining dataset is time-consuming. Sentence-T5 embeddings are 768-dimensional. When comparing one downstream example with all of C4, this can be calculated as a matrix-vector product between a matrix of dimension $365,049,127 \times 768$ and a 768-dimensional vector, followed by taking the maximum over the resulting vector. We paral-

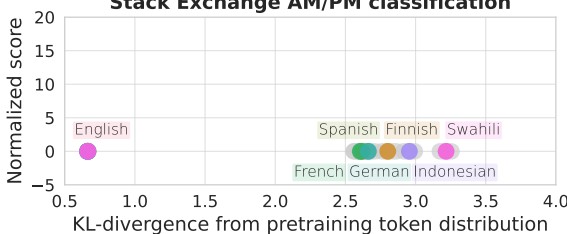

Figure 8: The same Stack Exchange datasets from Figure 1 are much more difficult when predicting whether the post occurred in the morning or afternoon. These datasets have nearly the same similarity to the pretraining dataset as their counterparts but are more difficult.

lelized this computation using a distributed SQL server and achieved runtimes of 90 seconds per example when comparing with the Pile and 74 seconds per example when comparing with C4. All other similarity metric runtimes were either less than one minute or calculated as part of evaluation.

## F   Additional figures

Figures 6, 7, and 8 give additional results referred to in the main paper.