# OpenReview forum: "Data Similarity is Not Enough to Explain Language Model Performance"
_EMNLP/2023/Conference — EMNLP 2023 Main_

### Official Review · Reviewer_pNy4 · 2023-08-03

**Soundness:** 4

**Excitement:**

4: Strong: This paper deepens the understanding of some phenomenon or lowers the barriers to an existing research direction.

**Missing References:**

A related area of reseach that isn't discussed in the paper is on task data leakage, or the presense of task examples in the pretraining dataset. This is a very specific case of data similarity where the similarity is exact match. Some examples of work in this area are [1], which audits the presense of task data in C4 (and is already cited in this work), and [2], which discusses the effect of task data presense during pretraining on downstream performance. Many pretraining papers also discuss this in their work (e.g., the GPT-2 and GPT-3 papers).

[1] Dodge et al. 2021. Documenting large webtext corpora: A case study on the colossal clean crawled corpus. In EMNLP.

[2] Magar and Schwartz. 2022. Data Contamination: From Memorization to Exploitation. In ACL.

**Paper Topic And Main Contributions:**

This paper asks whether (semantic) similarity to the pretraining corpus is correlated with model performance and tests this at both the task and example level. Specifically, the paper considers the overlap of both coarse-grained similarity (different languages) and finer-grain similarity within English benchmarks -- text classification, (X)NLI, and BigBench tasks -- against two open-source pretraining corpora: the Pile and C4; it tests whether these similarities are correlated with the performance of models (Pythia-6.9B and T5 3B) pretrained on these corpora. The experiments show that (a) for the multilingual evaluation, similarity to the pretraining data is correlated with downstream performance on that language and (b) this correlation does not hold for the finer-grained similarity evaluations within English. They also show that the different similarity metrics considered are not correlated with one another.

The primary contribution of this paper is that it tests the theory from in-context learning research that similarity to the pretaining data leads to better performance on that task and demonstrates that this is not always the case.

**Questions For The Authors:**

[A] Is there a reason why the MAUVE scores aren't included in the comparison in Table 2?

[B] It is interesting that in Table 1, the best correlations come from the best-performing model on KL-divergence and MAUVE. How do you think increasing model scale (and in theory model performance) will affect these results? I think it would be helpful to add a discussion of this to the limitations/discussion section.

[C] Line 58: Could you define the phrase "out-of-model embedding similarity"?

[D] The multilingual XNLI dataset covers 15 languages. Why did you only evaluate on five of these languages (including English), and why did you choose these five?

**Reasons To Accept:**

- The paper presents an interesting research question that addresses a common assumption made in LLM research (which is that similarity to the pretraining data explains why models perform well in-context on some tasks/examples but not others).
- The experiments presented in the paper are generally sound, well-thought-out, and support the claims made in the paper.
- The findings in the paper are interesting for the field and conflict with the intuitions held about how in-context learning works, which suggests we should further investigate why commonly-used text similarity measures do not relate to downstream performance.

**Reasons To Reject:**

I see no major reason to reject this paper. However, there are some minor issues that would strengthen the work if addressed:

- The discussion of LM generalization to different languages is inconsistent with the prior work in the area. For example, [1] shows that multiple "English" pretraining datasets do contain non-English text, which allows models trained on them to perform tasks outside of English; this paper specifically finds that this is true for C4. The other pretraining dataset considered, the Pile, is also estimated to be 2.6% non-English text [2, Section 5.2]. [3] also shows that GPT-3 performs well multilingually in some cases, despite not being explicitly multilingual. It would be helpful to rewrite the claims about multilinguality in LMs and their pretraining corpora to be more consistent with findings in this area.

- It would also be useful to flesh out the discussion of why the chosen similarity measures are not correlated with each other. The text could also expand on what each metric measuring in the main body (and to move Table 2 to the main body of the paper to accompany this discussion).

[1] Blevins and Zettlemoyer. 2022. Language Contamination Helps Explain the Cross-lingual Capabilities of English Pretrained Models. In EMNLP.

[2] Gao et al. 2020. The Pile: An 800GB dataset of diverse text for language modeling. arXiv.

[3] Lin et al. 2022. Few-shot Learning with Multilingual Language Models. In EMNLP.

**Reproducibility:**

4: Could mostly reproduce the results, but there may be some variation because of sample variance or minor variations in their interpretation of the protocol or method.

**Reviewer Confidence:**

5: Positive that my evaluation is correct. I read the paper very carefully and I am very familiar with related work.

---

> ### Author Rebuttal · Authors · 2023-08-28
>
> Thank you for your review! We really appreciate your thorough and constructive feedback.
>
> ***Concern:** The discussion of languages other than English is inconsistent with prior work. ‘English’ pretraining datasets contain some non-English text, and some language models perform well multilingually despite being trained on ‘English’ datasets.*
>
> Thank you for pointing this out and for pointing us to the references. We will update the text to state this and engage with these references. We will also state in the paper that the goal of the multilingual experiments in Section 3 is to use datasets that are more and less similar in aggregate to the pretraining datasets, which the x-axis in Figure 2 shows is the case.
>
>
> ***Concern:** Task data leakage is not addressed.*
>
> Thank you for directing us to this line of work. We will add it to our introduction and related work sections, from the perspective that leakage is an extreme case of item-level similarity. We will state in Section 4 how the ‘max cosine similarity’ metric is intended to explore this–downstream examples are compared to all pretraining documents, so exact matches will have a maximum cosine similarity of 1. We will highlight in the results how even high max cosine similarity does not guarantee high performance.
>
>
> ***Suggestion:** Further discuss why similarity measures are not correlated with each other in the main paper.*
>
> We will move Table 2 to the main paper and significantly expand our discussion using the extra page. We will also explain in more detail what each similarity is measuring at the beginning of Section 4.
>
>
> ***Question A:***
>
> Thanks for pointing out that comparisons with MAUVE score are missing from Table 2. This was an oversight, and we will add them in the final version. None of the correlations with MAUVE are significant.
>
> ***Question B:***
>
> We will expand on this observation in the results section and add to future work the question of whether there is a model scale threshold at which similarity does determine performance. We hypothesize that our results will be robust to model performance improvements since we study models with a range of performances across many different tasks, and we will add this to the discussion/limitations sections. At least some level of performance is necessary to see if similarity matters. For example, a model that performs poorly on all tasks (or conversely if it solves all tasks perfectly) will not be able to show any meaningful influence of similarity.
>
> ***Question C:***
>
> By ‘out-of-model embedding similarity’ we mean that texts are represented using embeddings from a model that is not necessarily the model used for prediction. We then calculate cosine similarity using these embeddings. We will replace this description in the paper.
>
> ***Question D:***
>
> We chose to evaluate languages from XNLI that are tokenized well by the Pythia tokenizer. In practice, this meant the five languages that use the Latin alphabet with few diacritics. The Pythia tokenizer is sufficiently targeted at the Latin alphabet that it breaks words from languages in most other scripts into sub-character-level byte tokens. We will state the selection criteria in Section 3 and discuss this limitation in the Limitations section. At your suggestion, we have run additional experiments on all languages in XNLI, and the results are nearly identical to those in Figure 2. We will include a figure in the appendix that shows all languages.
>
>
> ***Reproducibility:** The paper can only be reproduced with difficulty.*
>
> We will release code for reproducing our experiments, an initial version of which is currently available in the supplementary zip file. We will also release our raw similarities and performances.

---

### Official Review · Reviewer_ra9x · 2023-08-11

**Soundness:** 4

**Excitement:**

3: Ambivalent: It has merits (e.g., it reports state-of-the-art results, the idea is nice), but there are key weaknesses (e.g., it describes incremental work), and it can significantly benefit from another round of revision. However, I won't object to accepting it if my co-reviewers champion it.

**Paper Topic And Main Contributions:**

This paper investigates whether data similarity alone explains language model (LM) performance on different tasks. It challenges the common belief that tasks similar to LM's pretraining data should perform better. The paper contributes by:

1. Questioning the Similarity Hypothesis: Testing and disproving the idea that data similarity is the main determinant of LM performance.

2. Correlation Analysis: Evaluating various similarity measures and their correlation with LM performance on different benchmarks.

3. Benchmarking and Generalization: Assessing LM generalization across tasks and languages using diverse benchmarks.

4. Exploring Task Difficulty: Showing that similarity doesn't consistently predict easier examples.

5. Advocating Model-Based Interpretability: Highlighting the importance of alternatives to data similarity for explaining LM performance.

**Reasons To Accept:**

Strengths of the Paper:

1. Innovative Challenge: The paper questions the widely held "similarity hypothesis," adding a fresh perspective to the NLP community.

2. Thorough Evaluation: Rigorous analysis using diverse similarity metrics, datasets, and benchmarks enhances the study's robustness.

3. Well-Designed Experiments: Controlled scenarios with varied similarity conditions provide valuable insights into the complex data-performance relationship.

4. Surprising Insights: Counterintuitive findings highlight that data similarity might not solely dictate performance, sparking re-evaluation.

5. Broad Applicability: Accepted publication would prompt researchers to rethink language model design and fine-tuning strategies.

6. Engagement with Trends: Aligns with ongoing discussions about model interpretability, pretraining data, and generalization in NLP.


Benefits to the NLP Community:

1. Challenging Assumptions: Encourages a critical rethinking of established beliefs about data similarity and performance.

2. Enhanced Model Design: Insights can guide more robust model development, leading to better fine-tuning strategies.

3. New Research Paths: Sparks exploration into training data attribution and causal relationships, advancing the field.

4. Improved Interpretability: Promotes model-based interpretability approaches, enhancing model transparency.

5. Community Dialogue: Ignites valuable discussions and collaborations around the nuanced data-performance relationship.

6. Progressive NLP Advancement: Adds to evolving NLP research by questioning norms and pushing boundaries.

**Reasons To Reject:**

Potential Weaknesses:

1. Limited Scope of Datasets: The paper focuses on specific benchmarks and tasks, which might limit the generalizability of the findings. The results might not hold true for other tasks or datasets.

2. Simplistic Similarity Measures: The paper uses several measures of similarity, but they might not fully capture the complexity of the relationship between pretraining and task data. More nuanced measures could provide a more accurate representation.

3. Assumption of Linearity: The paper assumes that similarity is a linear predictor of performance, which might not be the case for all tasks and models. Non-linear relationships could influence the findings.

4. Complexity of Task Difficulty: The paper considers task difficulty as a constant factor, which might oversimplify its complexity. Task difficulty can be influenced by multiple factors beyond data similarity.

Potential Risks and Concerns:

1. Misinterpretation of Results: There's a risk that the findings could be misinterpreted, leading to the incorrect assumption that data similarity doesn't matter at all for LM performance. The paper should emphasize that while the correlation is not consistent across all tasks, it doesn't negate the potential influence of data similarity in certain contexts.

2. Critique of Methodology: Reviewers might question the validity of the chosen similarity measures, dataset selection, and the specific LM models used. There's a risk that the methodology might be deemed insufficient to definitively disprove the similarity hypothesis.

3. Controversial Conclusions: The paper's conclusion that "similarity doesn't explain performance" might generate controversy and skepticism within the NLP community. It's essential to carefully frame the conclusions to avoid misleading or polarizing interpretations.

4. Impact on Future Research Direction: The paper's findings might discourage researchers from exploring the relationship between data similarity and LM performance further. Authors should emphasize that the results warrant deeper investigation into more diverse contexts and tasks.

**Reproducibility:**

4: Could mostly reproduce the results, but there may be some variation because of sample variance or minor variations in their interpretation of the protocol or method.

**Reviewer Confidence:**

5: Positive that my evaluation is correct. I read the paper very carefully and I am very familiar with related work.

---

> ### Author Rebuttal · Authors · 2023-08-28
>
> Thank you for your review! We really appreciate your thorough engagement with our work. We believe we can address most of your concerns in the text of the paper, and we will include a new results table to address your concern about assuming linearity.
>
> ***Concern:** The paper’s conclusion that “similarity doesn’t explain performance” might be too broad, could be misinterpreted, and might prematurely discourage others from investigating further.*
>
> We share your concerns, and we will directly try to prevent any misleading interpretations by adding a paragraph to the discussion about what we are not claiming. For example, we will state that we are not claiming that similarity cannot matter or that the similarity metrics we evaluate capture all possible similarities. We do not definitively disprove the similarity hypothesis, but we do present evidence that the kinds of similarity that we consider are not all that matter for performance in this setting.
>
> We will more carefully frame the introduction, discussion, and conclusion to focus on the specific datasets and kinds of similarity metrics we consider. We will add open questions to the conclusion, like 1) whether more complicated metrics might capture the impact of similarity, 2) whether our results hold for more datasets, and 3) whether there is a model size threshold at which similarity does determine performance. We will also take your suggestion to “emphasize that while the correlation is not consistent across all tasks, it doesn't negate the potential influence of data similarity in certain contexts.” We definitely do not want to discourage more research on data similarity! We find that the impact of similarity is not as obvious as current literature makes it seem, and we hope future work can study more metrics, models, and tasks. Thank you for raising this concern.
>
> ***Concern:** Lack of linear correlation does not mean that similarity is unrelated to performance.*
>
> We agree and will state this in the paper. We have also performed Spearman rank correlation between similarity and performance, which measures the degree to which variables are related by a (possibly nonlinear) monotonic function. None of these correlations are significant after Bonferroni correction for multiple comparisons. We will discuss these results in Section 5 and add them to the appendix in a table that is similar to Table 1.
>
> ***Concern:** We are making a strong claim about similarity based on a limited set of metrics and tasks, and the results might not hold in other settings. The similarity measures are simple and may not capture the true relationship between pretraining and task data.*
>
> We will state that our results should be read in the context of these specific datasets and similarity measures. We agree that our results will benefit from future work that studies the same questions for additional datasets and more complex similarity metrics, and we will highlight such steps in our future work section. Although each similarity metric is relatively simple, we chose to study these because they are used in practice today. If similarity is so hard to pin down, then our results make it increasingly unlikely that similarity broadly construed is what determines performance in this setting.
>
> ***Concern:** Task difficulty is not a constant factor–difficulty can be influenced by many things, not just data similarity.*
>
> This is a great point! This is one of our main takeaways, so it is helpful to hear that it needs to be more explicit. We will devote more of Section 6 to this discussion, and we will explicitly state in the conclusion that there are many contributing factors to difficulty. We will also directly state that the results paragraph that starts in line 212 (with results in Figures 3, 4, and 5) controls for task difficulty by comparing individual examples within a dataset. One goal of this paper is to broaden the discussion of task difficulty beyond similarity.
>
> ***Reproducibility:** The paper can only be reproduced with difficulty.*
>
> We will release code for reproducing our experiments, an initial version of which is currently available in the supplementary zip file. We will also release our raw similarities and performances to facilitate further analyses.

---

### Official Review · Reviewer_tQTb · 2023-08-11

**Soundness:** 4

**Excitement:**

4: Strong: This paper deepens the understanding of some phenomenon or lowers the barriers to an existing research direction.

**Justification For Ethical Concerns:**

Please add an "Ethics Statement" section, although I do not see any ethical concerns.

**Paper Topic And Main Contributions:**

This paper studies the effects of the similarity between pretraining data and finetuning data on downstream model performance. The authors show that, using several similarity metrics, performance may *not always* be correlated with the similarity between the pretraining and finetuning data.

**Questions For The Authors:**

I would like to take this opportunity to congratualte the authors on a very nice piece of concise and thoughtful work.

**Reasons To Accept:**

This paper is very well written and tackles an important, underexplored problem and provides surprising new insights. In particular, it shows that similarity is not a one-size-fits-all explanation to model performance and exposes faults in this way of reasoning about model performance. This work provides valuable new insights to the community and should be accepted as is.

**Reasons To Reject:**

This paper is strong and runs more experiments than I would have expected from a short paper. I do not see any reasons to reject it.

**Reproducibility:**

4: Could mostly reproduce the results, but there may be some variation because of sample variance or minor variations in their interpretation of the protocol or method.

**Reviewer Confidence:**

4: Quite sure. I tried to check the important points carefully. It's unlikely, though conceivable, that I missed something that should affect my ratings.

**Typos Grammar Style And Presentation Improvements:**

Very well written. With only one grammatical point to make:
1. Lines 61 to 64 the part of the sentence in which you say "where we are able to change the similarity to an English pretraining dataset while keeping the underlying task fixed" is rather difficult to parse. I would consider rewording it or splitting it into two sentences.

---

> ### Author Rebuttal · Authors · 2023-08-28
>
> Thank you for your review! We really appreciate your positive and constructive feedback.
>
> Thanks for pointing out that lines 61-64 are confusing. We will rephrase the sentence to say: “First, we show that similarity correlates with performance for multilingual datasets. The same underlying task is represented in different languages, each with a different aggregate similarity to the pretraining dataset.”
>
> We will also add an ethics statement that discusses working with multilingual datasets and the role of computation in this research.

---

### Meta-Review · Area_Chair_c3mG · 2023-09-18

**Recommendation:** 5

**Metareview:**

This paper studies the effects of the similarity between pretraining data and finetuning data on downstream model performance. They find that the downstream performance may not correlate with the data similarities. The findings are interesting and insightful to the research community and experiments are thorough and extensive. Reviewers raised some concerns regarding the phrasing and discussions in the paper. Authors addressed these concerns very well. Thus I recommend accepting this paper to main conference.

---

### Decision · Program_Chairs · 2023-10-07

**Decision:**

Accept-Main

**Comment:**

This paper studies the effects of the similarity between pretraining data and finetuning data on downstream model performance. They find that the downstream performance may not correlate with the data similarities. The findings are interesting and insightful to the research community and experiments are thorough and extensive. Reviewers raised some concerns regarding the phrasing and discussions in the paper. Authors addressed these concerns very well. Thus I recommend accepting this paper to main conference.